# Physiological Benefits of Novel Selenium Delivery via Nanoparticles

**DOI:** 10.3390/ijms24076068

**Published:** 2023-03-23

**Authors:** Alice Au, Albaraa Mojadadi, Jia-Ying Shao, Gulfam Ahmad, Paul K. Witting

**Affiliations:** 1Redox Biology Group, Charles Perkins Centre, Faculty of Medicine and Health, School of Medical Sciences, The University of Sydney, Sydney, NSW 2006, Australia; 2Andrology Department, Royal Women’s and Children’s Pathology, Carlton, VIC 3053, Australia

**Keywords:** nano-selenium, growth, bioavailability, fertility, oxidative stress, inflammation

## Abstract

Dietary selenium (Se) intake within the physiological range is critical to maintain various biological functions, including antioxidant defence, redox homeostasis, growth, reproduction, immunity, and thyroid hormone production. Chemical forms of dietary Se are diverse, including organic Se (selenomethionine, selenocysteine, and selenium-methyl-selenocysteine) and inorganic Se (selenate and selenite). Previous studies have largely investigated and compared the health impacts of dietary Se on agricultural stock and humans, where dietary Se has shown various benefits, including enhanced growth performance, immune functions, and nutritional quality of meats, with reduced oxidative stress and inflammation, and finally enhanced thyroid health and fertility in humans. The emergence of nanoparticles presents a novel and innovative technology. Notably, Se in the form of nanoparticles (SeNPs) has lower toxicity, higher bioavailability, lower excretion in animals, and is linked to more powerful and superior biological activities (at a comparable Se dose) than traditional chemical forms of dietary Se. As a result, the development of tailored SeNPs for their use in intensive agriculture and as candidate for therapeutic drugs for human pathologies is now being actively explored. This review highlights the biological impacts of SeNPs on growth and reproductive performances, their role in modulating heat and oxidative stress and inflammation and the varying modes of synthesis of SeNPs.

## 1. Introduction of Selenium

Elemental selenium (Se) is an essential element with a capacity to ameliorate various biological functions, including antioxidant defence, redox homeostasis, growth, reproduction, immunity, and thyroid hormone production [1]. Selenium is sequestered primarily through dietary means and is present in trace amounts in the body although the distribution varies between tissue types. The recommended dietary dose exhibits a narrow margin with both beneficial and toxic effects documented [2,3]. Previous studies have shown that physiological levels outside the recommended range of Se intake are harmful; low dietary Se is linked to thyroid diseases, diabetes, and metabolic disorders while excessive Se causes cytotoxicity [4,5,6]. Therefore, tight regulation of optimal physiological Se levels is key for metabolic homeostasis and pharmacological safety. Dietary Se is generally obtained from seafood, organs, muscular meat cuts, grains, and seeds [7]. However, differing individual Se intakes due to local variations in Se soil content (including documented Se deficiency) has prompted the use of Se supplements to achieve the required daily Se intake [8]. Furthermore, as public health recommendations vary widely and local foods may also vary in elemental Se levels, the development of Se-based supplementation requires careful evaluation to ensure safe intake before advice can be publicised to human populations.

### 1.1. Selenoproteins

Biologically, Se is incorporated primarily into selenoproteins as the selenomethionine and selenocysteine amino acids at active sites [9]. For example, glutathione peroxidase (GPx) and iodothyronine deiodinase (DIO) are notable selenoproteins involved in antioxidant defence and thyroid hormone metabolism, respectively (refer to Figure 1). Current literature suggests that an adequate level of Se in the body is functionally important for various aspects, including protection against enhanced oxidative stress and alterations to physiological metabolic homeostasis. In contrast, individuals with a low Se status may adversely impact selenoprotein activity, where under severe conditions, Se deficiency and high Se status is coincident with a range of pathologies, including obesity [10,11], cancer [12,13], arthropathy [14] as well as several immune- and neurological-related disorders [15,16].

There is a growing body of research with a focus on the role of Se in cancer prevention which indicates a link between high Se exposure and decreased risk of breast, oesophagus, and prostate cancers [12,18,19]. Moreover, selenite has shown therapeutic effects against cancer, where selenite treatment stimulated cancer cell apoptosis via a mechanism involving enhanced ROS generation and the accumulation of hydrogen peroxide which, ultimately, decreased cell viability [20]. However, some studies show inconsistent results, where no beneficial effects were determined in intervention studies that increased Se intake in pathologies, including cancer, diabetes, and cardiovascular disease; moreover, Se supplementation correlated with increased occurrences of prostate cancer and diabetes [21,22].

Associations between Se and several other inflammatory disorders have been evaluated. For example, recent research demonstrated the anti-inflammatory role of Se in the millet-derived selenylated soluble dietary fibre [22] and nanoparticle formulation in a mouse model of inflammatory bowel disease (IBD) [23,24,25]. With respect to the latter, dietary Se was demonstrated to maintain intestinal microbiota homeostasis and barrier function in mice with DSS-induced colitis, with Se-mediated colon protective activity characterised by reduced ROS formation, enhanced antioxidant capacity, and dampened intestinal immune response, thereby protecting cellular functions and mitochondrial structures from excessive oxidative stress. Other observational studies also showed that IBD patients had lower Se levels than that of healthy individuals, potentially illustrating the importance of dietary Se as an antioxidant/micronutrient in the pathogenesis of IBD in humans [26,27].

Further research has demonstrated that Se status was associated with obesity, a chronic disease linked to inflammation, where either neutral or beneficial effects were observed with supplementation [11,28,29]. Interestingly, the available data indicate that sexual dimorphism plays a role in determining disease outcome; here lower Se status was linked to a higher BMI and increased risk of diabetes and reported incidences of myocardial infarction in females, yet the opposite was determined in males [30]. Conversely, contemporary research found that Se supplementation did not modulate oxidative damage and cellular proliferation within breast tissues in patients with an increased risk of breast cancer [31]. Other studies have also shown that patients with inflammatory diseases, including rheumatoid arthritis (RA), were commonly consuming a relatively high Se intake [32,33]. Yet other studies that investigated Se serum status correlated RA with lower Se levels in serum, suggesting that a potential mechanism of action involved the redistribution of Se serum to tissues to attenuate local oxidative stress and ROS production in RA pathogenesis [34,35].

The inconsistency in outcomes documented above is a complex issue due to inherent biological and environmental variations, along with the varying (baseline) Se status of individuals prior to supplementation, and the chemical forms and dosages used in reported Se supplementation studies. Nonetheless, dietary Se supplementation remains a popular option used to promote healthier lifestyles and optimise dietary habits, as enrichment with this micronutrient could potentially promote health benefits in humans [22,36].

### 1.2. Chemical Forms of Selenium

Dietary Se is available in different chemical forms. Although commercial Se supplements are widely available and easily accessible, the chemical forms of Se and the dosage across different products are inconsistent and not standardised. Furthermore, the chemical forms of dietary Se found in food are also diverse, including organic Se (selenomethionine, selenocysteine, and selenium-methyl-selenocysteine) and inorganic Se (selenate and selenite) of which the organic forms derived from plants are the main dietary source of Se in the human diet [37]. Therefore, intake of different Se-rich food sources and Se supplements might have distinct effects on metabolic pathways and show different biological and toxicological impacts on living systems [38,39,40]. Despite this knowledge gap, there is a lack of research comparing Se speciation in different chemical forms and their respective performance on various health aspects in humans, underscoring the need to determine an optimal chemical form of Se with maximal health benefits that can be endorsed and publicised in dietary recommendations.

Previous studies have largely investigated and compared the health impacts of different chemical forms of Se on farmed animals, such as poultry. The bioavailability of organic and inorganic forms of Se differs, with the retention of organic forms being higher and utilised by the body more efficiently than that of inorganic forms [41]. This differential retention of organo-Se forms has been ascribed to organic Se, such as selenomethionine, being chemically similar to methionine, which is commonly incorporated into the methionine pool within muscle as a form of Se storage. Therefore, on this basis selenomethionine is commonly used in feed for intensive poultry practices [6,41]. However, the inorganic forms are readily absorbed and in general the available evidence indicates inorganic Se displays enhanced protective effects against certain pathologies, including Parkinson’s disease [42,43]. Thus, sodium selenite has also ameliorated intestinal inflammation and histological damage in colitis mouse models via downregulation of proinflammatory cytokines and certain T-cell populations, with a concomitant upregulation of anti-inflammatory cytokines [44]. In parallel, selenite has shown to selectively promote apoptosis in prostate and lung cancer cells in vitro [20,45] suggesting another beneficial activity of this trace element when administered in this chemical form.

Recently, a raft of synthetic Se compounds with biological activities have been developed and their therapeutic effects have been studied in animal models that mimic mental disorders and degenerative diseases. For example, the selenocompound 3-[(4-chlorophenyl)selanyl]-1-methyl-1H-indole (CMI) has been shown to ameliorate blood–brain-barrier disturbances and diminish inflammation and oxidative stress in the brain of postseptic mice with psychological disturbances, including depression, anxiety, and cognitive impairment [46]. In another study, CMI was demonstrated to reverse stress-induced depression-like behavioural alterations, neuroinflammation, and oxidative imbalance in mice through a mechanism linked to reduction in corticosterone levels, enhanced antioxidant activities, and inhibited oxidative stress [47]. Similar therapeutic effects against depression-like behaviour and cognitive impairment induced by disease and treatment distress due to breast cancer have been observed [48]. Furthermore, CMI improved the inability to feel pleasure and anxiety induced by corticosterone through improved behavioural and biochemical alterations [49]. With the rising prevalence of depression and anxiety combined with mounting evidence that indicate Se’s anti-inflammatory and antioxidant properties, selenium-containing molecules are now considered a promising therapeutic candidate to potentially combat behavioural and biochemical alterations which are central to the development of psychological symptoms.

The delivery of bioactive cargo via nanoparticles represents a novel, emerging, and innovative technology. As an example, the application of nanotechnology in the delivery of minerals in animal-feed-enhanced absorption and nutrient bioavailability and has since revolutionised the poultry industry [50]. The available literature has shown that Se in the form of nanoparticles (SeNPs) has reduced toxicity, higher bioavailability, lower excretion, and is commonly linked to enhanced biological activities compared to the corresponding inorganic or organic selenocompounds in animals [51,52,53,54,55]. Therefore, this review highlights the biological impacts of SeNPs on growth and reproductive performances, its role in modulating heat and oxidative stress and inflammation, and the varying modes of synthesis of SeNPs. Although there have been several large-scale observational studies for various inorganic and organic Se chemical forms as mentioned previously, to the best of our knowledge there have been no trials using SeNPs in humans other than in vitro studies using human cell lines.

## 2. Selenium Nanoparticles

Despite the significance of Se on the physiological metabolic function, including growth as aforementioned, physiological levels of Se present with a narrow concentration range between nutritionally deficient, essential, and toxic doses [2]. Therefore, tight regulation of Se levels within a physiologically optimal range is critical for maintaining homeostasis and avoiding selenotoxicity. A major benefit of SeNPs is the significantly lower toxicity while retaining similar physiological impacts and efficacy in enhancing selenoprotein activities in comparison to that of other chemical seleno-forms as documented in some [3,55,56], though not all [57,58], of the available literature. This enhanced activity may be ascribed to the targeted delivery of Se to specific tissues. Despite the limited studies on SeNP toxicology, a study that investigated and compared the toxicology profiles of SeNPs with that of organic and inorganic Se demonstrated the significantly reduced risk of toxicity for the nano-vehicle form of Se [59]. Thus, mice administered with 2 mg SeNPs/kg body weight per day did not show evidence of suppressed growth unlike those mice administered organic and inorganic Se at the same dose, with selenite causing the most damage to the liver and kidneys. Selenium nanoparticles also caused less bone marrow cell death than other forms of Se and in addition, prevented DNA damage.

As synthesising Se using chemical methods could potentially employ toxic and harmful reagents, there is an increasing interest in the development and production of SeNPs using processes termed “green synthesis”, which employs microorganisms, including bacteria, fungi, and viruses to generate SeNPs, thereby minimising environmental pollution and potential health issues arising from chemical precursors [60].

The potential for SeNPs to be used as therapeutic agents is now being investigated with emerging evidence supporting a therapeutic advantage in diseases, such as Alzheimer’s disease, hepatic injury, and antimicrobial resistance [61,62,63]. These activities may be associated with SeNP’s improvement of drug delivery by enhanced selectivity and differentiation between diseased and healthy cells, thereby allowing for targeted release of the cargo in specific tissues and therefore, reducing the side effects [64].

In addition to improving diseased states, SeNPs are also being investigated in agriculture and food crop production. Presently, the accumulated evidence indicates that SeNPs represent a promising biological agent, where enhanced animal growth [65,66], improved feed conversion ratio [41,67], enhanced immunity (which imparts increased resistance to diseases and heat stress) [51,56,65,68,69], improved fertility [70,71], and preservation of meat quality [56,72] for human consumption have all been demonstrated in the aquaculture and poultry industries [52]. However, SeNP supplementation remains a challenge as Se supplementation with SeNPs is a relatively new concept and SeNPs needs to be manufactured at different concentrations to suit the dietary requirements of a wide range of animals with different baseline Se status to ultimately achieve optimal dietary supplementation. Further research is essential to establish supplementary dietary regimes for animal nutrition to produce Se-enriched meats and therefore, increase Se content in food for human consumption.

### 2.1. Mechanisms of Bioactivity for Selenium Nanoparticles

Despite the promising results with SeNPs, there is limited knowledge of mechanisms of SeNP absorption and metabolic conversion in the body, which has led to concerns surrounding the potential long-term toxicity of SeNP use [52]. Selenium nanoparticles show an enhanced uptake post-ingestion as these SeNPs are smaller in size with larger surface areas and are more permeable through capillary walls, leading to superior epithelial cell uptake and enhanced bioactivity. Notably, SeNPs exhibit lower rates of excretion compared to other forms of Se [73]. These combined properties of elevated uptake and decreased excretion facilitate the accumulation of SeNPs in the breast and duodenum tissues, due to the formation of nanoemulsion droplets with spherical and noncrystalline structures, with a lower accumulation in the liver involved in detoxification and clearance [74,75]. The same study also showed that SeNPs were nontoxic which was demonstrated by the absence of histological abnormalities in liver and brain tissues and paralleled findings from another study demonstrating significantly lower toxicity of SeNPs compared to other Se forms [56].

### 2.2. Impacts of Selenium Nanoparticles on Fertility

Infertility is a multifactorial disease that has increased among young males and females worldwide during recent decades [76]. Contemporary evidence demonstrates that reproductive disorders are commonly associated with genetic, lifestyle, and environmental factors, including dietary habits, use of recreational drugs, alcohol and caffeine consumption, and exposure to environmental pollutants during natural ageing [77,78] (refer to Figure 2). Chronic diseases, including diabetes, are often characterised by an unregulated production of ROS and consequential oxidative damage linked to DNA fragmentation, cytotoxicity, and death of sperm cells. This contributes to infertility, which could be attenuated by SeNPs in mouse models through a mechanism that includes the inhibition of lipid peroxidation and subsequent DNA damage [79]. Another contributing factor is that females in developed countries tend to delay childbearing and conception beyond peak fertility, even though the notion that human fertility declines with increasing age is well established [77].

Ample research has linked reproductive disorders and infertility to low Se status/Se deficiency while the outcome of assisted reproductive technologies (ARTs) in animals and humans with appropriate Se serum levels were associated with positive outcomes in conception, suggesting that an appropriate level of Se is essential for reproductive health [65,80,81,82]. This knowledge has directed contemporary research interests into exploring Se supplementation in both experimental animal models and in animal farming practice to enhance reproductive abilities. Notably, a significant body of this type of research has been focused on Se in the form of SeNPs due to its high bioavailability and relatively low toxicity compared to other Se forms (as mentioned previously) and therefore, this section of the review details the use of SeNPs in fertility research.

In vitro fertilisation (IVF) is a commonly used technique in various animal husbandry industries, such as the poultry industry, to enhance the desirable genetic qualities and increase the number of offspring. One of the main challenges for successful IVF in animals is the failure of oocyte maturation—a crucial step prior to fertilisation, due to oxygen exposure that manifests as enhanced oocyte oxidative stress [71,83]. Studies have demonstrated that supplementation with SeNPs enhanced the maturation rate in oocytes in vitro via upregulation of GPx4 and superoxide dismutatse (SOD) antioxidants, where the effect was more prominent in oocytes treated with SeNPs of 40 nm compared to that of 67 nm due to the relatively larger surface area and higher degree of cellular internalisation of the smaller SeNPs particles. Interestingly, SeNPs also improved the pluripotency and oocyte reprogramming, as marked by the upregulation of the developmental competence gene [71]. In further support of the potential bioactivity for vehicular delivered Se, in vitro bovine oocyte maturation, oocyte DNA integrity, and GSH concentration, characterised by increased re-expansion rate of blastocytes in vitro, were all demonstrated postsupplementation with 1 µg/mL of each of SeNPs or nano-zinc oxide [83]. Collectively, these reports underscore the significance of SeNPs in processes linked directly to in vitro maturation and fertilisation through a mechanism of enhanced antioxidant capacity and decreased oxidative stress.

Another key step in IVF for both humans and animals is semen cryopreservation—a process that necessarily involves repeated freezing and thawing cycles, which manifests the production of damaging ROS and causes oxidative stress which together reduces sperm viability by more than 35% [84]. Under these conditions, studies have shown that SeNP supplementation markedly improves gamete quality during IVF procedures [83,85]. For example, bovine semen supplemented with 0.5 and, 1.0 μg/mL of SeNPs prior to cryopreservation recorded enhanced post-thawing sperm motility and membrane integrity with reduced DNA damage and importantly, a higher fertility rate than that of the control [70]. These protective effects were attributed to the improved total antioxidant capacity in seminal plasma accompanied by decreased malondialdehyde concentration (MDA; a common marker of oxidative damage to polyunsaturated fatty acids of cell membranes) in the presence of supplemented Se. Therefore, the addition of an SeNP supplement to semen/ovum extenders in IVF protocols likely enhances the antioxidant status and preserves the quality of gametes and ultimately improves fertility rates.

As mentioned previously, chronic exposure to environmental pollutants could lead to permanent and irreversible damage to reproductive systems through induction of oxidative stress, DNA damage, loss of cell viability, and increased apoptosis, all of which could be attenuated by Se [86]. For example, aflatoxin, an environmental toxin found in the muscles and liver due to bioaccumulation from consuming agricultural products, including milk and eggs, has been attributed to inducing procarcinogenic and immunosuppressive effects [87]. A recent study showed that SeNPs attenuated testicular injury in aflatoxin B-exposed male mice due to its role in increasing the capacity of scavenging free radicals and diminishing the burden of ROS, thereby protecting spermatozoa and testis from lipid peroxidation and apoptosis [88]. The bioactivity of SeNPs also manifested as improved embryo production when sperm of aflatoxin B-exposed male mice was fused with healthy oocytes during in vitro fertilisation [88]. Similarly, another study showed that SeNPs rescued nickel-induced necrosis in seminiferous tubules via an upregulation of GPx enzymatic activity and downregulation of proapoptotic factors which inhibited caspase-mediated apoptosis, demonstrating the antioxidant potential and protective role of SeNPs against testicular injury induced by metal environmental pollutants [89]. In recent years, there has been an increased focus on combination therapies combining cisplatin with antioxidants to combat various side effects, including testicular dysfunction linked to infertility. Interestingly, SeNP administration improved the histological features and weight of testes in the presence of cisplatin-induced testicular toxicity [90]; once again indicating the potential for SeNPs to preserve viable sperm.

Another major reason for infertility is the presence of industrial chemicals in many consumer products that are endocrine disruptors, where long-term exposure to such chemicals is associated with reproductive dysfunction [91,92]. For example, bisphenol A (BPA) is an environmental toxin associated with plastics and has been linked to promoting infertility [77]. However, coadministration of Se to mice exposed to BPA improved the antioxidant activity and decreased ER-2 expression of gene involved in modulating apoptosis during spermatogenesis, thereby rescuing BPA-induced testicular damage and toxicity [93]. In fact, the protective effects of SeNPs surpassed that of supplemented (inorganic) sodium selenite as demonstrated by the ability to decrease proapoptotic mechanisms, DNA fragmentation, and selected gene expression levels, offering a potent strategy for reproductive protection against exogenous toxins.

Another common endocrine disruptor linked with reproductive toxicity and extensively used in consumer goods is di*-n-*butyl phthalate (DBP). Pregnant female rats exposed to DBP and administered with SeNPs gave birth to male offspring with enhanced testosterone levels, improved INSL3 and MR genes linked to Leydig cell functionality, improved antioxidant capacity, and reduced levels of MDA through the inhibition of lipid peroxidation, compared to those animals exposed to DBP alone [76]. Therefore, SeNPs could be a potential protective supplement used against reproductive toxicity induced by environmental toxins in the general population especially during the critical stages of pregnancy.

### 2.3. Impacts of Selenium Nanoparticles on Growth

It is a well-established fact that nutrition intervention affects the growth and fertility performance and antioxidant status of the progeny [50]. Importantly, the increasing demand for fish and meat as primary dietary source and the pursuit for high-quality foods with advantageous health features have instigated new research in nutritional enhancement to supply the world with an improved quality of food at higher production rates [72]. Recently, attention has been focused on supplementation with SeNPs as one of the primary dietary approaches used in the aquaculture, agriculture, and poultry industries [94].

Fish is a primary source of animal protein in many countries [95]. However, the rising consumer demand and increase in environmental pollution have led to challenges in aquaculture and nutritional research which is essential for a sustainable supply of fish and future food security globally [96]. Nile tilapia, rich in omega-3 fatty acids, is one of the most widely consumed fish worldwide [97]. Studies investigating supplementation of Nile tilapia with 1 mg SeNPs /kg body weight have demonstrated improved growth performance as measured by a greater weight gain compared to that of the control; however, this beneficial action of SeNPs was limited, as fish supplemented with a dose above or below 1 mg/kg showed no enhanced weight gain [72]. In addition, fish supplemented with 1 mg SeNPs /kg body weight exhibited a desirable fatty acid profile and simultaneous enhancement to antioxidant potential as demonstrated by a higher polyunsaturated fatty acid content and a significantly increased GPx activity, respectively. These outcomes further elucidate the importance of the dose of Se used in animal feed and the growth-promoting potential of SeNPs and the results were also corroborated in intensive poultry farming [65]. Another similar study compared the effects of SeNPs to that of the traditional Se forms (organic and inorganic) using Nile tilapia [51]. Outcomes from this study indicated that fish supplemented with SeNPs showed the optimal haematological profile using measurements of haemoglobin, red blood cells, and circulating IgM levels, suggesting improved health and immunological status when using this form of dietary Se. Furthermore, SeNPs simultaneously improved SOD, catalase (CAT), and GPx activities and reduced MDA levels in the liver, suggesting enhanced hepatic antioxidant defence which was consistent with a previous study [72]. Additionally, SeNPs improved intestinal health, which was demonstrated by a greater villus length and the number of goblet cells in the colon mucosa indicating more efficient digestion and utilisation of food and enhanced mucosal protection for the underlying intestine.

Importantly, research has also show that dietary Se improved the gut microbiome favourably by enhancing the abundance of beneficial bacteria and limiting the growth of undesirable pathogens [98]. In comparison with other metallic nanoparticles, such as silver and gold nanoparticles with antimicrobial properties, SeNPs are relatively less toxic as they are intrinsically essential for metabolic processes and are present in the biological system [99,100]. Broiler birds fed with 0.9 mg SeNPs /kg body weight impacted the gut bacterium genus and enhanced intestinal health, demonstrating an increased abundance of beneficial bacteria, including *Lactobacillus* and *Faecalibacterium*; a phenotype that is linked to favourable metabolite production, including butyrate [101]. Under these same dietary supplementation conditions, the production of short-chain fatty acids was also enhanced, and this was associated with improved immunity and colonic mucosal function, with commensurate reduction in the risk of inflammation, diabetes, and IBD. Taken together, these results suggested the potential application of SeNPs in poultry feed to produce beneficial health outcomes via the modification of the gut microbiota.

Global warming causes a rise in sea temperature which negatively impacts growth, metabolism, and various physiological functions of aquatic animals, and ultimately reduces survival rates. Under these conditions, the mechanism involved increases free radical production and reduces tissue damage which could be attenuated by SeNP supplementation [102]. Furthermore, a recent study found that SeNP supplementation alleviated heat stress and improved thermal tolerance in rainbow trout through upregulation of GPx and CAT activities and activation of glutamate-glutamine pathways linked to reduced ROS production and inflammation [103]. Microscopically, the phospholipid membrane of rainbow trout was more integral due to SeNP-induced mitigation of oxidative damage. Another similar study showed that SeNPs promoted protein repair and inhibited apoptosis in rainbow trout via upregulation of heat stress proteins and downregulation of proapoptotic proteins and cholesterol synthesis [104].

In the agriculture industry, SeNPs ameliorated heat stress in underdeveloped piglets, characterised by enhanced plasma SOD, CAT, and GPx activities, increased anti-inflammatory IL-10 cytokine, and decreased levels of the oxidation biomarker MDA [68]. Another recent study established the role of SeNPs in improving resistance to biological stress in plants with the aim of limiting the use of chemical pesticides where potential toxicity in humans was an inherent risk [105]. Furthermore, SeNPs improved resistance to pathogen invasion in melon plants via increased SOD and CAT activities and their mRNA levels along with increased APX and POD activities suggesting improved antioxidant capacity and ROS scavenging [106]. Enhanced photosynthesis in melons was accompanied by the detection of increased abundance of mitochondria, chlorophyll, and demonstrable thickening of cell walls. Taken together, these results demonstrated the significance of SeNPs in mediating tolerance of heat stress via enhanced antioxidant capacity and thermal and biological stress tolerance, thereby improving the viability of biological organisms.

Numerous studies have documented that maternal dietary intervention with SeNP supplementation affects the growth and fertility performance and antioxidant status of the progeny [52,107]. Compared to sodium selenite and selenium yeast, SeNPs were most effective in enhancing egg production, egg weight, and feed conversion ratio in laying hens, accompanied by significantly increased Se concentration in eggs, GPx1 liver mRNA levels and serum GPx activity, and lower MDA levels [66]. Interestingly, SeNPs enhanced the tolerance of laying hens to deoxynivalenol (DON; a fungal toxin) most likely through a mechanism of SeNP-mediated enhancement of antioxidant activity [65]. Thus, laying hens exposed to DON and fed with SeNP supplemented feed demonstrated increased antioxidant defence and immune response to DON, characterised by improved GPx levels, higher egg production, and protection against oxidative damage and soft-shelled or cracked egg rates.

Although a suitable amount of heat is key in embryonic development and growth, high incubation temperatures could adversely impact hatchability and growth performance of broilers [65]. Yet, in ovo injection of SeNPs during late incubation in broilers significantly enhanced antioxidant capacity and attenuated oxidative stress, which manifested as decreased cortisol levels and an increased T3/T4 ratio, suggesting reduced heat stress and increased thyroid hormone metabolism. Again, these results reinforce the importance and potential application of SeNP supplementation to enhance food production and quality in the poultry industry.

### 2.4. Impacts of Selenium Nanoparticles on Diseases and Human Health

As aforementioned, Se possesses anti-inflammatory and antioxidant properties. Thus, a body of research has investigated the effects of Se specifically in the form of SeNPs on alleviating diseases, including diabetic nephropathy [108], Alzheimer’s disease [61], and leukemia [109] in vitro and in animal models.

The most common chronic diseases linked to natural aging, including diabetes, Alzheimer’s disease (AD), and cardiovascular disease, and cancers are characterised by increased ROS production, oxidative stress, and chronic inflammation. In diabetic rats, SeNPs have shown to reduce proinflammatory markers, including IL-1β and TNF levels and renal MDA levels leading to lower oxidative stress, indicated by improved renal functions due to lower serum urea and creatinine along with reduced glucose level [108]. Furthermore, resveratrol (RSV) a naturally occurring chemical found in grapes with neuroprotective properties showed maximal therapeutic effects against AD when delivered to rats in an RSV-SeNPs cocktail [110]. In terms of biological mechanisms of action, the synergistic interaction of RSV and SeNPs could better attenuate lipid peroxidation, alleviate mitochondrial membrane disruptions, and restore the levels of antioxidant enzymes in AD-affected brain tissues. Rats with AD administered with the RSV-SeNPs cocktail displayed improved AD symptoms, shown by improved acetylcholine levels and disrupted formation of Aβ aggregates accompanied by enhanced clearance of Aβ peptides, thereby inhibiting the local inflammatory responses.

The potential therapeutic benefit of SeNP activity against oxidative stress and inflammation has also been demonstrated in a model of cardiac oxidative damage and fibrosis induced by the low thyroid hormone levels in hypothyroid rats [111]. Here, SeNPs administered at 150 µg/kg attenuated cardiac fibrosis and hypertrophy of cardiomyocytes via enhanced CAT and SOD activities and thiol levels, with dampened MDA levels in cardiac tissues. Similarly in a model for vascular endothelial cell dysfunction and injury in rats induced by homocysteine, SeNPs, sodium selenite, and selenomethionine improved the vascular phenotype through the rescue of local GPx1 and GPx4 levels [112]. Notably, under these conditions SeNPs exhibited lower toxicity compared to other Se forms while maintaining a similar level of vaso-protection. The lower toxicity of SeNPs was corroborated in another study where the LD_50_ of SeNPs was 18-fold more than that of selenite in mouse models, with less Se retention [113]. Nevertheless, SeNPs when administered at the same dose as selenite not only reduced the levels of TBARS (another secondary marker of lipid peroxidation), but also elevated GSH levels.

In addition to SeNP’s therapeutic properties in chronic diseases, vehicular transport of Se is reported to display either synergistic effects with cancer drugs or show anticancer activity specific to cancer cells, suggesting less toxicity and collateral damage to healthy cells and tissues. For example, SeNPs promoted swelling and cell lysis in acute myeloid leukaemia (AML) cells via cell cycle arrest and apoptosis, while displaying minimal toxicity to haematopoietic stem cells and T cells [109]. Notably, the SeNPs used in this study were embedded in nanotubes consisting of triple helix β-d-glucan (BFP) polysaccharide extracted from black fungus which encouraged adhesion to biological tissues, consequently increasing BFP-SeNP absorption and retention rates. Similarly, SeNPs encapsulated in gold nanocages were released into target tissues upon radiation and stimulated local apoptosis due to ROS-induced mitochondrial dysfunction, when used alongside with a cancer drug, suggesting the potential targeted release of SeNPs [64]. Importantly, SeNPs enhanced the elimination efficacy of cancer cells compared to normal healthy cells which was confirmed by another study [114]. Furthermore, the enhancement of selective apoptosis using SeNPs has also been demonstrated in breast cancer, under experimental culture conditions where SeNPs were added to breast cancer cells in vitro 24 h prior to irradiation treatment [115].

To enhance the selectivity of SeNPs in a cancer-targeted drug delivery system, research has shown that SeNPs attached to folate (FA) prior to loading the cancer drug ruthenium polypyridyl (RuPOP), effectively enhanced the drug specificity so that RuPOP was only released in an acidic microenvironment (e.g., the stomach) to facilitate drug release in an on-demand fashion [116]. Altogether, these results suggest the potential application of SeNPs as a synergistic cancer treatment due to its selectivity in promoting cancer cell death, likely through SeNPs’ activation on cell cycle arrest at the G2/M phase, metabolic stress, and increased intracellular ROS production in cancer cells. In addition to the apoptotic activity of SeNPs against cancer cells, SeNPs conjugated to quercetin (Qu) and acetylcholine (ACh) presenting antibacterial activity against multidrug-resistant superbugs (MDRs) by causing irreversible damage to the bacterial cell wall upon adhesion [63].

As mentioned previously, SeNPs could attenuate the negative impacts of environmental toxins and chemicals commonly present in commercial products on fertility, and their beneficial effects could also be demonstrated in other diseases. Selenium nanoparticles could restore neurotoxicity and motor deficits in rats induced with cypermethrin (CYP) pesticide by mitigating the oxidative stress due to CYP metabolism in the liver which yields ROS and oxidative stress [117]. Neurotoxic mice treated with SeNPs displayed normal behavioural outcomes which was linked to the increased levels of GABA and glutathione and lower levels of MDA and inflammatory markers (TNF-α and IL-1β), thereby preventing the excessive CYP-induced excitation of the neuronal system. In addition, SeNPs protected the liver and kidneys against the toxic effects induced by a widely used analgesic drug, acetaminophen, via the maintenance of DNA integrity and improved hepatic antioxidant capacity as supported histologically by reduced hepatic oxidative lesions and restored hepatic cellular structure [62].

Interestingly, a recent study investigated the potential analgesic effects of SeNPs in inflammatory disorders, as inflammation promotes the release of inflammatory mediators which excite nociceptive neurons leading to local inflammation and pain [118]. Despite SeNPs’ anti-inflammatory activity supported by reduced leukocyte numbers and proinflammatory cytokines, including prostaglandin, TBAR and NOx markers, SeNPs had no impact on the nociceptive threshold in rat models.

### 2.5. Synthesis of Selenium Nanoparticles

The daily requirement of Se intake that the body needs can be obtained by eating Se-enriched foods, including vegetables, grains, and meat; yet the daily requirement of Se might not be sufficiently met through dietary consumption alone [119]. Compared with the conventional Se supplementation forms available in the market, SeNPs outperform the organic and inorganic Se forms in terms of bioactivity and toxicity. In light of the potential application of SeNPs in the aquaculture, poultry, and human supplement industries, SeNPs could become a novel form of Se supplementation [76]. There are several pathways to synthesise SeNPs, of which the most common types are chemical synthesis which uses various reagents, and biosynthesis involving plants or microorganisms to produce the encapsulated trace element [120]. For the chemical methodology, preparation methods of SeNPs and the characterisation methods of both synthetic methodologies are generally standardised, however the preparation for the biosynthesis of SeNPs varies widely depending on the type of plants and microorganisms used, which are summarised and discussed herein.

For the preparation of SeNPs by chemical synthesis, the standard protocol utilised by most studies involved the addition of 1 mL of 25 mM NaSe solution into 4 mL of glutathione (GSH) of the same concentration containing 15 mg of bovine serum albumin (BSA), following the supplementation of a stabilising polymer and pH adjustment to 7.2 [41,117]. It is worth noting that the size and surface charge of SeNPs could be manipulated by altering the pH of the solution and the BSA amount used in the manufacturing process [3,121]. The final solution containing the SeNP product was purified by dialysis against double-distilled water for 4 days, and the water was replaced daily to isolate the by-product GSH from the final product. The morphology and identity of the final product was characterised and verified using X-ray diffraction, placing a small sample stained with phosphotungstic acid (2%) on a copper grid and viewing the sample under transmission electron microscopy (TEM). Another preparation method employed selenium dioxide (SeO_2_) dissolved in distilled water containing 0.2% polyvinylpyrrolidone (PVP) to obtain selenious acid solution, which turned clear upon cooling in an ice bath [118]. The addition of an ice-cold 0.1 mol/L reducing agent, potassium borohydride, initiated a colour change in the SeO_2_ solution from clear to a yellow–orange, which was indicative of the formation of SeNPs. Similar to the characterisation method mentioned previously, the resultant solution containing SeNPs was then characterised using dynamic light scattering (DLS) and electrophoretic light scattering (ELS) techniques to determine the zeta potential value and provide the morphology of the SeNPs. Higher zeta potential, regardless of the sign (i.e., +/−) is indicative of particle stability and this manifests as an increased resistance for aggregation. This is explained by a lower value for the zeta potential, meaning that attractive forces might exceed the interparticle repulsion and the dispersion might break and form small clumps/masses. Therefore, an appropriate zeta potential value is critical to the formation and stability of SeNPs [122].

For the biosynthesis of SeNPs, several studies employed a technique similar to the chemical syntheses (described above) but instead used plants and ascorbic acid as a chemical reductant [76,113,123]. Water-soluble and natural polymer polysaccharides, such as chitosan, konjac glucomannan, acacia gum, carboxymethyl cellulose, glucan derived from plants, such as black fungus [109] and Lentinus Edodes (shiitake mushroom) [114], and plant roots, including Withania Somnifera [79], were employed, as they are excellent stabilisers in the synthesis of dispersed colloidal SeNPs [123]. Selenium nanoparticles manufactured using this protocol used seleniuous acid solution in a mixture containing polysaccharides, such as cellulose, yielding a Se/cellulose aqueous suspension which was mixed with ascorbic acid. The ascorbic acid solution was slowly added into the Se/cellulose suspension and vigorously stirred until the suspension changed colour from white to brick red/orange as the Se/cellulose colloids and SeNPs began to form. The final product could then be separated, purified by dialysis, and washed with water and 70% ethanol to remove excess ascorbic acid and other low molecular weight by-products, then dried using a spray-drying process to remove residual moisture.

Another similar study investigating the antimicrobial actions of SeNPs against superbugs utilised quercetin (Qu), an essential plant flavonoid present in many fruits, flowers, and vegetables due to its inherent antibacterial properties, and also an acetylcholine (ACh) neurotransmitter due to its ability to combine with the receptor present on bacteria cell wall, thereby promoting the binding of Qu-ACh-SeNPs to bacteria [63]. To synthesise Qu-ACh-SeNPs, NaSe was mixed with Qu dissolved in methanol and ACh chloride in the presence of acetic acid for 10 min in cold conditions. Following the addition of a reducing agent, sodium borohydride, the mixture was vigorously stirred. The final red solution was centrifuged to collect the red precipitate, followed by repeated washing with PBS to obtain the final complex Qu-ACh-SeNP product.

Synthesis of SeNP can also utilise bacterial cell lysates [124] or anaerobic granular sludge containing bacteria to biologically reduce selenite forming biogenic SeNPs [3]. Steps for this protocol include gathering granular-sludge biofilms from an anaerobic sludge blanket reactor treating paper mill wastewater which can then be used in the microbial conversion of soluble Se oxyanions to insoluble elemental Se, which is incorporated into biogenic SeNPs as confirmed by the appearance of a red-coloured substance. Mechanistically, the bacteria synthesised SeNPs by either selenite denitrification involving nitrate reductases and cellular cofactors (present in E. coli), the reduction of Se (IV) compounds involving nitrite reductases (present in Rhizobium), and selenate reductase involved in the chemical reduction of selenate, or selenite reduction by reduced thiols, such as GSH (present in eukaryotic cells, cyanobacteria, and the α-, β- and γ-groups of the proteobacteria) [60]. This process has the added potential benefit of reducing the discharge of Se-contaminated sludge into the aquatic environment considering the increasing concerns around Se toxicity in aquatic animals [57]. Interestingly, biogenic SeNPs were found to be 3.2-fold less toxic than selenite and 10-fold less toxic than chemically synthesised SeNPs in zebrafish embryos when comparing LC_50_ values [3]. Despite the lower toxicity of biogenic SeNPs compared to that of the chemically synthesised SeNPs, one inherent risk associated with this mode of synthesis is that a substantial amount of biogenic SeNPs could remain in the bioreactor due to its colloidal structure resulting in discharge to the aquatic environment [3]. Nonetheless, biogenic SeNPs are less toxic with lower bioavailability due to the presence of extracellular polymeric substances, which decreases the interactions between nanoparticles and biological organisms. In contrast to using biological sludge in producing biogenic SeNPs, other studies have utilised fish organs, such as gills [57]. In these studies, homogenised fish gills were centrifuged to collect the supernatant, which was mixed with 200 mL of 2M sodium selenite and shaken continuously for 36 h. After centrifugation, the harvested pellets were dried and stored at room temperature so that they could later be crushed to form a gross colloid as required for dietary supplementation.

As mentioned previously, designated SeNPs could be used as a drug delivery system to enhance the selectivity of the cargo between diseased and healthy cells, where synthesis of SeNPs often involves the use of metal nanocages, including gold. This manufacturing process involves two steps, (i) the synthesis of gold nanocages, and (ii) the loading of selenious acid and phase change materials (PCM), such as lauric acid, into the nanocages [64]. Firstly, silver nanocubes of 45 mm are converted to gold nanocages (AuNCs) through galvanic replacement reaction with HAuCl_4_. Secondly, 1 mL AuNCs is dispersed in 0.5 mL of methanol containing 0.3 g of PCM (lauric acid) and 0.2 g of selenious acid, which is stirred at 50 °C for 5 h and then centrifuged at 14,000 rpm to obtain SeNPs encapsulated in gold nanocages which can then be redispersed in 1 mL of deionised water to obtain the final product. Similarly, another study prepared two types of selenium nanoparticles with quantum mechanical properties that showed growth inhibition on cancer cells through the induction of mitochondria-mediated apoptosis and necrosis [125]. Another study investigating the therapeutic enhancement effects of SeNPs for the cancer drug, ruthenium polypyridyl (RuPOP) conjugated folate (FA) and SeNPs to RuPOP, producing FA-RuPOP-SeNPs, which was synthesised in a similar fashion using ascorbic acid, sodium selenite, and polysaccharide polymers [116]. FA receptors are commonly overexpressed in cancer cells, which is a characteristic that could be leveraged when devising cancer drug treatments to facilitate the delivery and uptake of FA-RuPOP-SeNPs by cancer cells.

## 3. Conclusions

Dietary supplementation of SeNPs have demonstrated a range of benefits in agriculture and humans, with superior performance over traditional chemical forms of dietary Se. Therefore, it is increasingly seen as an innovative and novel alternative. Recognising the antioxidant properties of dietary Se, the benefits ascribed to SeNPs include lower toxicity, enhanced growth performance, improved nutritional quality, immune functions, improved overall reproductive performance, reduced oxidative stress and inflammation, ameliorated heat stress, neurotoxicity and environmental pollutants, increased resistance to infections, assistive effects in chemotherapy, selective apoptosis of cancer cells, selective delivery of cancer drugs, and antibacterial actions against superbugs. However, further studies are still required to elucidate pharmacological activity and safe dosages due to inherent biological and environmental variations, and the different (baseline) Se status for animals and humans in different regions of the world. The advantages mentioned above in combination with the multiple synthetic pathways used to develop tailored SeNPs for use in intensive agriculture, and as potential therapeutics for human pathologies, make this form of Se a prime candidate for further research. However, in addition to limited toxicity data, currently there is a paucity of complete pharmacokinetic data, including SeNP uptake vs. half-life and metabolism, and the accumulation of the SeNPs or a metabolite in tissue vs. dosage. Once a more complete pharmacokinetic study is available for optimal forms of SeNPs, further testing of these optimal Se delivery systems will require extensive future studies on humans using randomised control trials.

## Figures and Tables

**Figure 1 ijms-24-06068-f001:**
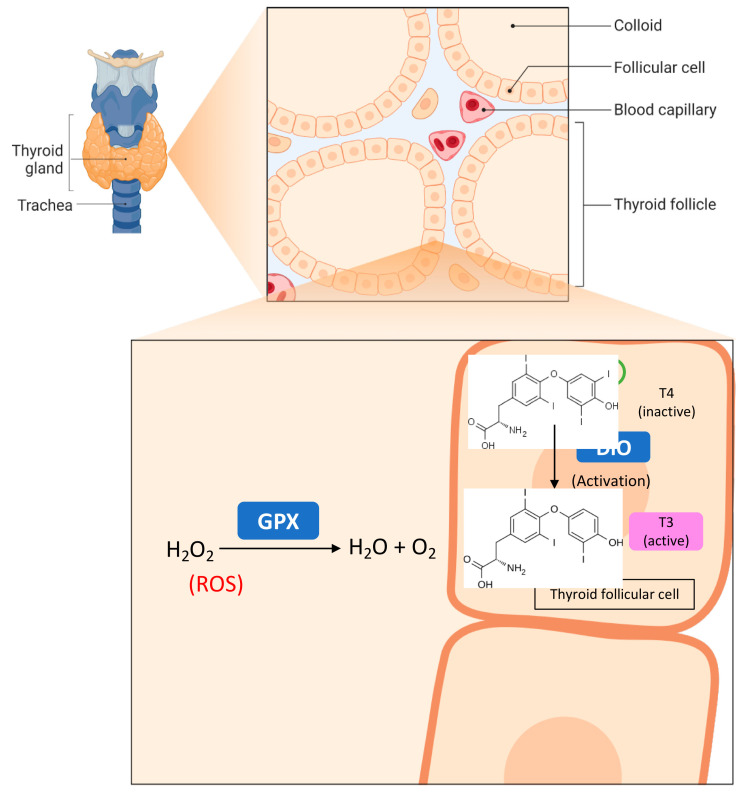
The thyroid gland and the role of selenoproteins in the thyroid (adapted from “Thyroid Gland Anatomy and Histology”, by BioRender.com (on 1 March 2023). Retrieved from https://app.biorender.com/biorender-templates, accessed on 17 January, 2023) [17]. Glutathione peroxidase (GPx) selenoprotein found in the colloid of the thyroid is involved in antioxidant defence. GPx degrades hydrogen peroxide, an endogenous reactive oxygen species (ROS) generated during normal thyroid hormone synthesis, thereby protecting the thyroid from excessive oxidative stress. Iodothyronine deiodinase (DIO) selenoprotein is present in thyroid follicular cells. DIO converts inactive thyroxine (T4) thyroid hormone into its biologically active triiodothyronine (T3) thyroid hormone to regulate downstream body metabolism in various tissue beds.

**Figure 2 ijms-24-06068-f002:**
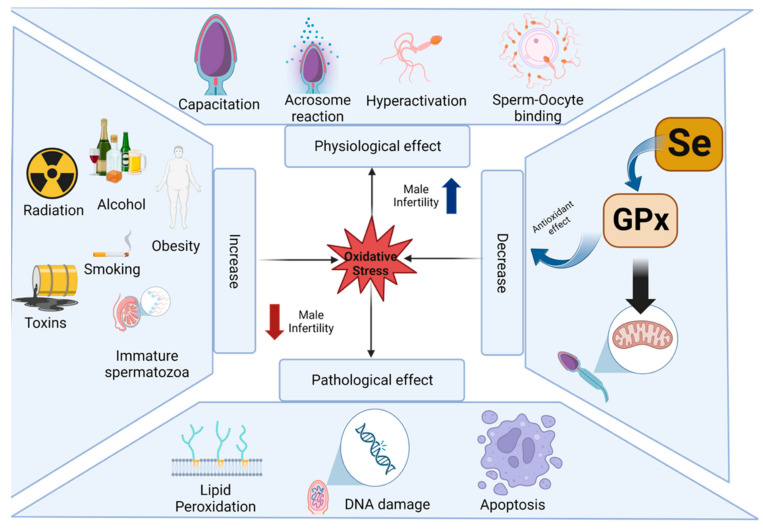
Schematic highlights various lifestyle factors that contribute to unfavourable physiological effects, including increased oxidative stress, and pathological conditions leading to male infertility in the long term (retrieved from https://app.biorender.com/biorender-templates, accessed on 17 January 2023). Poor lifestyle, including radiation, alcohol, smoking, exposure to toxins, and obesity could impact sperm capacitation, acrosome reaction, hyperactivation, and sperm-oocyte binding via molecular actions, including lipid peroxidation, DNA damage, and apoptosis, contributing to male infertility. Conversely, adequate Se intake can promote pathways leading to optimal activity of antioxidants, including GPx which alleviates oxidative stress involved in male infertility.

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
