# Peer review of "Physiological Benefits of Novel Selenium Delivery via Nanoparticles"

_ijms, 2023, doi:10.3390/ijms24076068_

Round 1

Reviewer 1 Report

The authors have done an excelent work in describing the field of selenium nanoparticles. The manuscript is well organized and illustrated. A few suggestions is described below:

1. In section 1.2, the authors should mention that synthetic compounds containing selenium are a big field of study, where many biologically active selenium-containing molecules have been developed. For reference, please cite: https://www.sciencedirect.com/science/article/pii/S0009279720309911 ; https://link.springer.com/article/10.1007/s00213-018-5151-x ; https://www.sciencedirect.com/science/article/pii/S0889159119312292 ; https://www.ingentaconnect.com/content/ben/cmc/pre-prints/content-35708081 ; https://www.sciencedirect.com/science/article/pii/S266649762100028X .

2. The authors should include a brief description about the toxicology of selenium nanoparticles. 

3. Have selenium nanoparticles been used in the field of veterinary medicine or agriculture? If so, I believe it is worth mentioning in the review, simply to make it as comprehensive as possible. 

4. In the conclusion, the authors should add a couple of sentences about the future directions of this field of selenium nanoparticles. 

Author Response

Reviewer 1

The authors have done an excellent work in describing the field of selenium nanoparticles. The manuscript is well organized and illustrated.

We thank the reviewer for these positive comments and address the issues raised as per outlined below.

A few suggestions are described below:

  1. In section 1.2, the authors should mention that synthetic compounds containing selenium are a big field of study, where many biologically active selenium-containing molecules have been developed. For reference, please cite: https://www.sciencedirect.com/science/article/pii/S0009279720309911 ; https://link.springer.com/article/10.1007/s00213-018-5151-x ; https://www.sciencedirect.com/science/article/pii/S0889159119312292 ; https://www.ingentaconnect.com/content/ben/cmc/pre-prints/content-35708081 ; https://www.sciencedirect.com/science/article/pii/S266649762100028X .
  • As suggested by the reviewer we have added new text at Line 139-158 as follows:

“Recently, a raft of synthetic Se compounds with biological activities have been developed and their therapeutic effects have been studied in animal models that mimic mental disorders and degenerative diseases. For example, the selenocompound 3-[(4-chlorophenyl)selanyl]-1-methyl-1H-indole (CMI) has been shown to ameliorate blood-brain-barrier disturbances and diminish inflammation and oxidative stress in the brain of post-septic mice with psychological disturbances, including depression, anxiety, and cognitive impairment [46]. In another study, CMI was demonstrated to reverse stress-induced depression-like behavioural alterations, neuroinflammation, and oxidative imbalance in mice through a mechanism linked to reduction in corticosterone levels, enhanced antioxidant activities and inhibited oxidative stress [47]. Similar therapeutic effects against depression-like behaviour and cognitive impairment induced by disease and treatment distress due to breast cancer have been observed [48]. Furthermore, CMI improved the inability to feel pleasure and anxiety induced by corticosterone through improved behavioural and biochemical alterations [49]. The rising prevalence of depression and anxiety combined with mounting evidence that indicate Se’s anti-inflammatory and antioxidant properties, selenium-containing molecules are now considered a promising therapeutic candidate to potentially combat behavioural and biochemical alterations that are central to the development of psychological symptoms.”

  1. The authors should include a brief description about the toxicology of selenium nanoparticles. 
  • As suggested by the reviewer we have added new text focusing on what is presently known about Se cytotoxicity at Line 32-36 as follows:

“Previous studies have shown that physiological levels outside the recommended range of Se intake are harmful; low dietary Se is linked to thyroid diseases, diabetes, and metabolic disorders while excessive Se causes cytotoxicity [4-6]. Therefore, tight regulation of optimal physiological Se levels is key for metabolic homeostasis and pharmacological safety.”

In addition, we note that we had previously mentioned this at Lines 173-177 (and again Toxicity specific to SeNPs in Line 180-188 as follows) however, as requested we have now explored this topic earlier as suggested by the reviewer. There is also limited studies regarding Se toxicity specific to SeNPs due to the field only now developing into nanltechnology.

  • “Despite the limited studies on SeNP toxicology, a study that investigated and compared the toxicology profiles of SeNPs with that of organic and inorganic Se demonstrated the significantly reduced risk of toxicity for the nano-vehicle form of Se [54]. Thus, mice administered with 2 mg SeNPs/kg body weight/per day did not show evidence of suppressed mice growth unlike those mice administered organic and inorganic Se at the same dose, with selenite causing the most damage to liver and kidney. Selenium nanoparticles also caused less bone marrow cell death than other forms of Se and in addition, prevented DNA damage.”
  1. Have selenium nanoparticles been used in the field of veterinary medicine or agriculture? If so, I believe it is worth mentioning in the review, simply to make it as comprehensive as possible.
  • Yes, we agree with the reviewer that this is an important topic. We have identified this in the original manuscript – please refer to a brief description between lines 200-206, as follows:

  • (i) In addition to improving diseased states, SeNPs are also being investigated in agriculture and food crop production. Presently, the accumulated evidence indicates that SeNPs represent a promising biological agent, where enhanced animal growth [60, 61], improved feed conversion ratio [41, 62], enhanced immunity (that imparts increased resistance to diseases and heat stress) [51, 60, 63-65], improved fertility [66, 67], and preservation of meat quality [63, 68] for human consumption were all demonstrated in aquaculture and poultry industries [52].
  • (ii) and then more extensively in Sections 2.2 and 2.3 in the revised manuscript.

  1. In the conclusion, the authors should add a couple of sentences about the future directions of this field of selenium nanoparticles.
  • We note, we have already mentioned “Further studies are still required to elucidate pharmacological activity and safe dosages due to inherent biological and environmental variations, and the different (baseline) Se status for animals and humans in different regions of the world.”
  • However, as suggested by the reviewer we have also added a section (refer to Line 632-637 in the revised manuscript) to further emphasise the need for further studies.

“However, in addition to toxicity data there is only a paucity of complete pharmacokinetic data – uptake vs half-life and metabolism, and accumulation of the SeNPs or a metabolite in tissue vs dosage. Once a complete pharmacokinetic study is available for optimal forms of SeNPs, further testing of the optimal Se-delivery system will require extensive studies in humans by way of future randomised trials.”

Reviewer 2 Report

The review is very well formatted and should be accepted post minor revisions-

1. Add more recent references and details on clinical trials conducted using selenium in the past

2. It's crucial to address the toxicity associated with selenium nanoparticles and their management approach

3. The limitations of current selenium delivery vehicles can be highlighted in the review.

Overall, the review looks impressive.

Author Response

Reviewer 2

The review is very well formatted and should be accepted post minor revisions-

We thank the reviewer for these positive comments and address the issues raised as per outlined below.

  1. Add more recent references and details on clinical trials conducted using selenium in the past
  • There are many observational studies for inorganic and organic Se (this has been reviewed previously by us in Line 48-100 and Line 130-138). However, to the best of our knowledge there have been no trials using SeNPs in humans other than using in vitro human cell lines (added in Line 167-170).
  1. It's crucial to address the toxicity associated with selenium nanoparticles and their management approach
  • As suggested by the reviewer 1 we have added new text focusing on what is presently known about Se toxicity at Line 32 as follows:
    • “Previous studies have shown that physiological levels outside the recommended range of Se intake are harmful; low dietary Se is linked to thyroid diseases, diabetes, and metabolic disorders while excessive Se causes cytotoxicity [4-6]. Therefore, tight regulation of optimal physiological Se levels is key for metabolic homeostasis and pharmacological safety”.
  • There are limited studies regarding Se toxicity specific to SeNPs, however Toxicity specific to SeNPs in Line 180-188 is as follows.
    • “Despite the limited studies on SeNP toxicology, a study that investigated and compared the toxicology profiles of SeNPs with that of organic and inorganic Se demonstrated the significantly reduced risk of toxicity for the nano-vehicle form of Se [54]. Thus, mice administered with 2 mg SeNPs/kg body weight/per day did not show evidence of suppressed mice growth unlike those mice administered organic and inorganic Se at the same dose, with selenite causing the most damage to liver and kidney. Selenium nanoparticles also caused less bone marrow cell death than other forms of Se and in addition, prevented DNA damage.”
  1. The limitations of current selenium delivery vehicles can be highlighted in the review.
  • What has been written in to the original manuscript includes sections of text related to:
    • The relatively poor cellular uptake which can be combatted by using biological synthesis of SeNPs rather than chemical synthesis  which has already been mentioned.
    • Issue sof environmental toxicity for chemical synthesis of Se miucronutrients  also mentioned.
    • Instability of SeNPs  also mentioned.

However, as suggested by the reviewer we have expanded slightly on this topic by identify that there is little data on SeNPs at present as this is a developing field.

  • Line 632-634: The only thing is the paucity of complete pharmacokinetic data – this goes beyond toxicity and looks at uptake vs half-life and metabolism, accumulation of the SeNP or a metabolite in tissue vs dosage….

Overall, the review looks impressive.

Reviewer 3 Report

This is a well-written review describing the biological impacts of selenium nanoparticles. Some of the references are not appropriately cited as shown below.

1. lines 74-75: Ref. 22 is not appropriate here because it used millet-derived selenylated soluble dietary fiber, which is not SeNPs.

2. lines 139-141, lines 150-152: Cite a reference(s) shown below for the properties of SeNPs.

1) H.L. Wang, J.S. Zhang, H.Q. Yu, Elemental selenium at nano size possesses lower toxicity without compromising the fundamental effect on selenoenzymes: comparison with selenomethionine in mice Free Radic. Biol. Med., 42 (2007), 1524-1533

2) J.S. Zhang, X.Y. Gao, L.D. Zhang, Y.P. Bao, Biological effects of a nano red elemental selenium, Biofactors, 15 (2001), 27-38

3) Zhang, J., Wang, H., Yan, X. and Zhang, L. Comparison of Short-Term Toxicity between Nano-Se and Selenite in Mice. Life Sciences, 76 (2005), 1099-1109

3. line3 150-152: Ref. 47 is not suitable here because the paper concluded as “In the present study, we found that Se-NPs have more toxicity than Se”. Related with this, the following paper also suggests toxicity of SeNPs to fish.

Li H, Zhang J, Wang T, Luo W, Zhou O, Jiang G (2008) Elemental selenium particles at nano-size (Nano-Se) are more toxic to Medaka (Oryzias latipes) as a consequence of hyper-accumulation of selenium: a comparison with sodium selenite. Aqua Toxicol 89:251–256

4. line 191: Citing ref. 47 is wrong as mentioned above.

5. lines 188-191: Again, ref. 47 suggests “Se-NPs have more toxicity than Se”.

6. line 139: Cite a reference for “lower excretion” of SeNPs.

7. lines 235-239: Ref. 71 shows the effects of CuO-NPs and Zn-NPs on oocytes, not SeNPs.

8. lines 245-246: Both ref. 71 and 72 do not describe about SeNPs.

9. lines 356-358: Cite ref. 57.

10. line 543-545: The sentence may misread because Ref. 48 describes that selenate reductase is involved in selenate reduction and that selenite is reduced by thiols such as GSH.

Author Response

Editorial comment:

This is a well-written review describing the biological impacts of selenium nanoparticles. Some of the references are not appropriately cited as shown below.

  1. lines 74-75: Ref. 22 is not appropriate here because it used millet-derived selenylated soluble dietary fiber, which is not SeNPs.

We thank the reviewer for picking this issue up and we have adjusted the text as follows:

  • For example, recent research demonstrated the anti-inflammatory role of Se in the dietary form millet-derived selenylated soluble fiber [22] and nanoparticle formulation in a mouse model of inflammatory bowel disease (IBD) [23-25].

  1. lines 139-141, lines 150-152: Cite a reference(s) shown below for the properties of SeNPs.

1) H.L. Wang, J.S. Zhang, H.Q. Yu, Elemental selenium at nano size possesses lower toxicity without compromising the fundamental effect on selenoenzymes: comparison with selenomethionine in mice Free Radic. Biol. Med., 42 (2007), 1524-1533

2) J.S. Zhang, X.Y. Gao, L.D. Zhang, Y.P. Bao, Biological effects of a nano red elemental selenium, Biofactors, 15 (2001), 27-38

3) Zhang, J., Wang, H., Yan, X. and Zhang, L. Comparison of Short-Term Toxicity between Nano-Se and Selenite in Mice. Life Sciences, 76 (2005), 1099-1109

We have adjusted the text as guided by the reviewer as follows (adding additional references [51-55]:

  • The available literature has shown that Se in the form of nanoparticles (SeNPs) has reduced toxicity, higher bioavailability, lower excretion, and is commonly linked to enhanced biological activities compared to the corresponding inorganic or organic selenocompounds in animals [51-55].

  1. line3 150-152: Ref. 47 is not suitable here because the paper concluded as “In the present study, we found that Se-NPs have more toxicity than Se”. Related with this, the following paper also suggests toxicity of SeNPs to fish.

Li H, Zhang J, Wang T, Luo W, Zhou O, Jiang G (2008) Elemental selenium particles at nano-size (Nano-Se) are more toxic to Medaka (Oryzias latipes) as a consequence of hyper-accumulation of selenium: a comparison with sodium selenite. Aqua Toxicol 89:251–256.

We have adjusted the text as guided by the reviewer (see below) - Here we added appropriate refs that document a lower toxicity for SeNP, but we also retained Ref 47 (now cited as Ref 57 in the revised manuscript R2) and new Ref 58 to provide a balanced viewpoint.

  • A major benefit of SeNPs is the significantly lower toxicity while retaining similar physiological impact and efficacy in enhancing selenoprotein activities in comparison to that of other chemical seleno-forms as documented in some [3, 55, 56], though not all of the available literature [57, 58]; this enhanced activity may be ascribed to the targeted delivery of Se to specific tissues.

  1. line 191: Citing ref. 47 is wrong as mentioned above.

We have adjusted the text as guided by the reviewer (see below) and removed the citation to Ref 47 (cited in the original revised manuscript R1) now Ref 57 (as cited in the revised manuscript R2) and instead cited an appropriate paper as shown below.

  • The same study also showed that SeNPs were non-toxic as judged by the absence of histological abnormalities in liver and brain tissues, which paralleled findings from another study that corroborated significantly lower toxicity of SeNPs compared to other Se forms [56].

  1. lines 188-191: Again, ref. 47 suggests “Se-NPs have more toxicity than Se”.

We have adjusted the text as guided by the reviewer (see below) and removed the citation to Ref 47 in the original revised manuscript R1 (now Ref 57 in the revised manuscript R2) and instead cited an appropriate paper.

  • Furthermore, SeNPs simultaneously improved SOD, catalase (CAT) and GPx activities and reduced MDA levels in the liver, suggesting enhanced hepatic antioxidant defense which was completely consistent with a previous study [72].

  1. line 139: Cite a reference for “lower excretion” of SeNPs.

We have adjusted the text as guided by the reviewer (see below).

  • Notably, SeNPs exhibit lower rates of excretion compared to other forms of Se [73].

  1. lines 235-239: Ref. 71 shows the effects of CuO-NPs and Zn-NPs on oocytes, not SeNPs

We have adjusted the text as guided by the reviewer (see below).  Notably, Ref 71 in the original revised manuscript R1 reports on nano zinc and nano selenium, not nano zinc and nano copper. Last line of Abstract: Optimal embryo development was partially dependent on the presence of NSe and NZn-O during IVM. NSe and NZn-O during oocyte maturation act as a good cryoprotective agents of vitrified, warmed blastocysts.

However, we have clarified this text as follows:

  • In further support of the potential bioactivity for vehicular delivered Se in vitro bovine oocyte maturation, oocyte DNA integrity, and GSH concentration, characterised by increased re-expansion rate of blastocytes in vitro, were all demonstrated post-supplementation with 1 µg/mL of each of SeNPs or nano-zinc oxide [83].

  1. lines 245-246: Both ref. 71 and 72 (now Refs 83 and 85 in the revised version R2) do not describe about SeNPs.

  • Note, Ref 71 in the original revised manuscript R1: Abdel-Halim, B.R. and N.A. Helmy, Effect of nano-selenium and nano-zinc particles during in vitro maturation on the developmental competence of bovine oocytes. Animal Production Science, 2018. 58(11): p. 2021-2028.

Note, Ref 72 in the original revised manuscript R1: Li, S., et al., Glutathione and selenium nanoparticles have a synergistic protective effect during cryopreservation of bull semen. Front Vet Sci, 2023. 10: p. 1093274.

Hence the text reads as shown below in the revised manuscript R2:

  • Under these conditions, studies have shown that SeNP supplementation markedly improves gamete quality during IVF procedures [83, 85]

  1. lines 356-358: Cite ref. 57.

We have adjusted the text as guided by the reviewer (see below) – note Ref 57 in the original manuscript R1 is now Ref 68 in this revised version R2.

  • In the agriculture industry, SeNPs ameliorated heat stress in underdeveloped piglets, characterised by enhanced plasma SOD, CAT and GPx activities, increased anti-inflammatory IL-10 cytokine and decreased MDA levels [68].

  1. line 543-545: The sentence may misread because Ref. 48 (in the original manuscript R1) describes that selenate reductase is involved in selenate reduction and that selenite is reduced by thiols such as GSH.

We have adjusted the text as guided by the reviewer (see below) and added additional text to clarify which reductant is involved in each case:

  • Mechanistically, the bacteria synthesised SeNPs by either selenite denitrification involving nitrate reductases and cellular cofactors (present in E. coli), reduction of Se (IV) compounds involving nitrite reductases (present in Rhizobium), and selenate reductase involved in the chemical reduction of selenate while selenite is reduced by thiols such as GSH (present in eukaryotic cells, cyanobacteria, and the α-, β- and γ-groups of the proteobacteria) [60].